# Comparison of Reconstituted, Acidified Reconstituted Milk or Acidified Fresh Milk on Growth Performance, Diarrhea Rate, and Hematological Parameters in Preweaning Dairy Calves

**DOI:** 10.3390/ani9100778

**Published:** 2019-10-10

**Authors:** Lingyan Li, Jiachen Qu, Xiaoyue Xin, Shuxin Yin, Yongli Qu

**Affiliations:** 1Key Laboratory of Efficient Utilization of Feed Resources and Nutrition Regulation in Cold Region of Heilongjiang Province, College of Animal Science and Veterinary Medicine, Heilongjiang Bayi Agricultural University, Daqing 163319, China; llytiger@163.com (L.L.); 15147012655@163.com (X.X.);; 2College of Animal Science and Technology, Inner Mongolia Agricultural University, Hohhot 010018, China; qjc990313@163.com

**Keywords:** reconstituted milk, acidified reconstituted milk, acidified fresh milk, growth performance, diarrhea rate, hematological parameters, calves

## Abstract

**Simple Summary:**

The preweaning phase is the period for the rapid growth and development of dairy calves. During this period, dairy calves receive their nutrients through milk. Feeding hygienic milk is of great benefit to optimum growth rate and health status of dairy calves. Previous studies focused on the effects of hygienic milk by acidification on dairy calves’ health and growth. Reconstituted milk, as the common source of milk, is being used in dairy calves feeding. However, no previous studies reported the effects of feeding acidified reconstituted milk on dairy calves’ health and growth. Our study will provide the evidence that the acidification of reconstituted milk had positive effects on growth performance and health status of preweaning dairy calves.

**Abstract:**

The present experiment was carried out to assess the effects of reconstituted milk (RM), acidified reconstituted milk (ARM), and acidified fresh milk (AFM) on growth performance, diarrhea rate, and hematological parameters of preweaning dairy calves. For this purpose, a total of 27 Holstein female calves (one month of age) with initial body weight of (67.46 ± 4.08) kg were divided into three groups in such a way that each group contained nine calves. Calves were housed individually, and starter was offered ad libitum to each calf. The dietary treatments were RM, ARM, and AFM. The highest milk intake was observed in calves receiving AFM as compared to other treatments (*p* < 0.01). Calves fed AFM had more feed intake than those fed ARM and RM (*p* < 0.01). Feed efficiency was significantly lower for calves offered ARM than those offered RM and AFM (*p* < 0.01). A lower withers height growth was found for calves fed RM than those fed ARM and AFM (*p* <0.05). Diarrhea rate and white blood cell (WBC) and lymphocytes (LYM) counts were greater for calves fed RM than those fed ARM and AFM (*p* < 0.05). These findings suggested that ARM and AFM had positive effects on growth performance and health status of the preweaning dairy calves.

## 1. Introduction

The preweaning phase is the period for the rapid growth and development of female dairy calves [1]. Providing calves with sufficient nutrients to achieve the optimum growth rate in preweaning phase is beneficial for promoting growth performance after adulthood, which positively associated with increased productivity in later life [2].

On large scale dairy farms, pre-weaning dairy calves are usually reared by milk or milk replacer [3,4,5]. Delay of feeding fresh milk to calves results in the multiplication of bacteria in the milk due to favorable conditions. Therefore, preservation of milk is recommended at large-scale dairy farms to overcome the consequences of bacterial load in fresh milk. It has been reported that acidification of milk to a pH value between 4.0 and 4.5 can inhibit pathogenic bacterial growth and enable milk to be preserved without refrigeration for a short time prior to feeding to calves [6,7]. It has also been reported that acidified milk protects calves from diseases caused by bacteria growth and reduces the incidence of diarrhea [8].

Reconstituted milk is resulted from the addition of water to dried or condensed milk powder in the amount necessary to reestablish the specified water:solids ratio. Reconstituted milk is also being used on the dairy farms for preweaning dairy calves [9]. However, to our knowledge, the acidification of reconstituted milk in preweaning growing calves has not been investigated. Previous studies only focused on acidified milk or acidified milk replacer in dairy calves [10,11,12]. Therefore, a comparative study was undertaken to compare the growth performance, diarrhea rate, and hematological parameters of preweaning dairy calves fed reconstituted milk, acidified reconstituted milk, or acidified fresh milk.

## 2. Materials and Methods

### 2.1. Calves, Feeds and Management

The animal care and experimental procedures were conducted by the Animal Welfare and Ethics Committee of Heilongjiang Bayi Agriculture University. Animal care and handling were followed the guidelines by the regulations for the Administration of Affairs Concerning Experimental Animals (The State Science and Technology Commission of China, 1988). Twenty-seven Holstein female calves procured from large dairy herd of same age with initial body weight of (67.46 ± 4.08) kg were selected and randomly divided into three groups in such a way that each group contained nine calves. Three types of milk were applied to different treatment groups—(1) reconstituted milk (RM), which was made by mixing milk powder with warm water (35–40 °C) in a weight ratio of 1:7; (2) acidified reconstituted milk (ARM), which was obtained by adding 30 mL formic acid (food grade) of 8.5% concentration in 1L reconstituted milk at temperature 5–10 °C; and (3) acidified milk made by using fresh milk (AFM), which was obtained by adding 30 mL formic acid (food grade) of 8.5% concentration in 1L fresh milk at temperature 5–10 °C. The fresh milk was procured from general and healthy herd and not pasteurized before feeding. Milk samples of 50 mL were collected every 10 days to detect the compositions. A milk analyzer (Foss Milkoscan 4000, Foss, Hillerod, Danmark) was used to determine the composition of milk, and milk composition are shown in Table 1. The pH value of acidified milk was determined by a pH meter (FE20, Mettler Toledo, Zurich, Switzerland). Milk samples of 50 mL were successively diluted (10^−1^ to 10^−4^) using distilled water, and a volume of 1 mL of each dilution was inoculated onto the surface of plate count agar (PCA), violet red bile agar (VRBA), and MRS agar for the cultivation of total bacteria, lactobacillus, and *Escherichia coli*, respectively. Total bacteria and *Escherichia coli* were incubated at 37 °C for 48 and 24 h. Lactobacillus was incubated anaerobically at 37 °C for 72 h. After incubation, the numbers of total bacteria, lactobacillus and Escherichia coli colony-forming units (cfu) were calculated. Bacterial numbers of different types of milk are shown in Table 2.

Prior to onset of trial, all the calves were offered the same basal ration from day 1 to 21. Experimental treatments were applied from day 22 onward. Calves were given a nine-day adaptation period before the start of data collection. No abnormal behaviors were observed during the adaptation period, and the initial data were collected on day 30. Calves were housed individually in calf hutches, and 6 L milk were offered twice daily at 5:00 and 16:30. All calves had free-choice access to clean, fresh water and starter throughout the experiment. The chemical compositions of the starter diets are shown in Table 3. The experiment period lasted for 40 days until weaning. Standard management and environmental conditions were ensured to avoid any stress as described in recent researches [13,14].

### 2.2. Growth Performance

Calves were weighed before the morning feeding for three consecutive days at the start and end of the experiment to calculate the initial body weight, final body weight, and average daily gain (ADG). Milk and starter consumption were recorded daily. Feed efficiency was determined from total feed consumption and ADG.(1)$$\mathrm{Feed}\;\mathrm{efficiency}\;=\;\frac{\mathrm{Total}\;\mathrm{feed}\;\mathrm{intake}(\mathrm{kg}/\mathrm{day}\;\mathrm{DM})}{\mathrm{ADG}(\mathrm{kg}/\mathrm{day})}$$

In addition, body measurements of each calf including body length, withers height, heart girth, and shin circumference were taken at the beginning and end of the trial according to the study made by Li et al. (2014) [15].

### 2.3. Fecal Score and Diarrhea Rate

Fecal consistency was scored everyday on a scale of 1 to 4, where 1 = normal-firm but not hard, 2 = soft-does not hold form, 3 = runny-spreads easily, and 4 = devoid of solid matter. Fecal score of 3 or 4 were considered as diarrhea. The diarrhea rate was calculated according to the procedure described in the recent study [16] and formula is given below.
(2)$$\mathrm{Diarrhea}\;\mathrm{rate}\;=\;\frac{\mathrm{number}\;\mathrm{of}\;\mathrm{calves}\;\mathrm{with}\;\mathrm{diarrhea}\;\times \;\mathrm{days}\;\mathrm{of}\;\mathrm{diarrhea}}{\mathrm{total}\;\mathrm{number}\;\mathrm{of}\;\mathrm{calves}\;\times \;\mathrm{examined}\;\mathrm{days}}\;\times \;100$$


### 2.4. Hematological Parameters

Blood samples (about 10 mL) were taken from the jugular vein in the morning, approximately 3h post feeding on the first and final day of the experiment. Blood was anticoagulated with EDTA for hematological analysis within three hours after collection. Hematological parameters values including white blood cell (WBC), lymphocytes (LYM), monocytes (MO), red blood cell (RBC), hemoglobin (Hb), and hematocrit (HCT) were detected by using automatic hematology analyzer (XN9000, Sysmex, Kobe, Japan).

### 2.5. Statistical Analysis 

All the data were analyzed by PROC MIXED of SAS version 9.4 (SAS Institute Inc., Cary, NC, USA), the statistical model was
Y_ij_ = μ + D_i_ + C_j_ + ε_ij_(3)
where Y_ij_ is the observed variable, μ is the overall mean, Di is the fixed effect of diet treatment, C_j_ is the random effect of calves, and ε_ij_ denotes the residual error. Significance was declared for *p* < 0.05, and trends were reported at 0.05 < *p* < 0.10. When a significant effect of treatment was detected (*p* < 0.05), differences between the means were tested using Bonferroni multiple comparison test.

## 3. Results

### 3.1. Growth Performance

The results of growth performance of dairy calves are shown in Table 4. The initial BW, final BW, ADG, and starter intake were not different among treatments. Milk fed amount in liters were not different among different treatments. However dry matter intake from milk differed among treatments. Lowest dry matter intake from milk was observed in calves fed ARM, while the highest dry matter intake from milk was observed in calves fed AFM (*p* < 0.01). Calves fed AFM had increased total feed intake compared to those fed ARM and RM (*p* < 0.01). The value of feed efficiency was significantly lower for calves offered ARM than those offered RM and AFM (*p* < 0.01), which suggested that the feed efficiency was better for calves fed ARM than RM and AFM.

Body measurements data of different treatments are presented in Table 5. Body measurements were not different at the beginning of the experiment. Smaller heart girth was observed in the end of the experiment for calves fed AFM than those fed ARM and RM (*p* < 0.05). Compared with calves fed ARM and AFM, a lower withers height growth could be found for calves fed RM (*p* < 0.05). Similarly, final withers height was lower for calves fed RM as compare to ARM and AFM (*p* < 0.05).

### 3.2. Fecal Score and Diarrhea Rate

Fecal score and diarrhea rate of dairy calves were presented in Table 6. Average fecal consistency score and diarrhea rate during 30–50 day, 51–60 day, and 61–70 day were greater for calves fed RM than those fed ARM and AFM (*p* < 0.01).

### 3.3. Hematological Parameters

Hematological parameters of different treatments are presented in Table 7. Hematological parameters values of WBC, LYM, MO, RBC, HB, and HCT were not different among the treatments at the start of the experiment. At the end of the trial, higher white blood cell and lymphocytes counts were found in calves offered RM (*p* < 0.05).

## 4. Discussion

### 4.1. Growth Performance

Preweaning growth of female dairy calves is considered an important factor of milk production in their later stage of life. Feed intake, ADG, and feed efficiency are important indicators for growth performance of female dairy calves. Feed intake affects the level of nutrient intake, thus affecting ADG and final BW. ADG and final BW play an important role in assessing the feed efficiency and overall health status of dairy female calves [17]. Akins reported that female diary calves should gain 0.75 kg per day to reach optimum weight of weaning at eight weeks of age [18]. In the present study, ADG of calves in different treatment groups exceeded 0.75 kg/day. Although ADG were in favor of calves fed ARM and AFM compared to those fed RM (0.92, 0.96 vs. 0.88 kg/day), however the differences were not significant. Eren found that the calves offered acidified milk had increased ADG than those offered whole milk [19]. Zhang et al. reported that acidified milk replacers contribute to the improvement of growth rate of calves [12]. Dry matter intakes (DMI) from milk was the highest for calves fed AFM. This was expected as the highest total solid concentration of AFM which resulted in the higher consumption of DM from milk by calves. DMI from starter were not different for calves fed ARM, RM, and AFM, which were 0.43, 0.40, and 0.42 kg/day DM, respectively. The results are consistent with other research [10,11,12,20]. However, Sun et al. found a reduction in the intake of starter for the calves fed acidified milk with butyric acid addition [16]. Interestingly, in the current study, starter feed DMI was similar among all the group, which could be explained by different acidifier and experimental treatments as reported in the previous studies. Overall, ARM, RM, and AFM behave similarly on the intake of starter feed in the current study.

Body measurements usually reflect body growth and development, which can be used to estimate liveweight [21]. Calves assigned to the ARM and AFM treatment groups exhibited greater withers height gain in the body measurements than the RM calves. The results suggested that calves reared using ARM and AFM had better skeletal growth. Todd et al. reported the calves fed acidified milk replacer were associated with greater preweaning structural growth [5]. However, some studies found that body measurements of calves were not affected by feeding acidified milk or acidified milk replacer [3,10,16].

### 4.2. Fecal Score and Diarrhea Rate

Diarrhea is the most important cause of disease of calves, with high morbidity and mortality, which threatens the health and growth performance of calves and is a major cause of economic loss for cattle raisers [22]. Results in the present study showed that calves fed ARM and AFM exhibited lower fecal consistency scores and diarrhea rate. It has been reported that lower pH values inhibit the growth of total bacteria and *Escherichia coli* [23], and the current study provided the evidence that ARM and AFM contained lower number of total bacteria and *Escherichia coli* (Table 2). The lower number of total bacteria and *Escherichia coli* reduces the growth and reproduction of saprophytic bacteria that harm the health of gastro-intestinal tract and cause diarrhea. On the other hand, feeding acidified milk can increase the acidity in the intestinal tract of calves, as reported by Woodford et al. [24]. Relatively higher acidity of intestinal tract is likely to exhibit a bacterial influence, therefore reducing the incidence of diarrhea among calves. Therefore, the lower incidence of diarrhea in ARM and AFM could be explained by the lower pH of intestine caused by ARM and AFM and by the lower number of total bacteria and *Escherichia coli*.

### 4.3. Hematological Parameters

Hematological parameters such as WBC, LYM, and RBC counts and Hb concentration are considered as important clinical indicators that are widely used to reflect health and disease status [25]. Diseases of the preweaning calves is one of the major cause of economic loss in dairy production [26]. Thus, the measurement of hematological parameters could be used to evaluate the health and physiological status of the calves [27]. WBC are the major part of the immune system that helps fight infection and defend the body against other foreign materials. Different types of WBC are involved in recognizing intruders, killing harmful bacteria, and creating antibodies to protect the body against future exposure to some bacteria and viruses [28,29]. LYM are one of several different types of white blood cells that are also important in the immune system, with T cells being responsible for directly killing many foreign invaders and B cells producing antibodies to guard the body against infection [30]. Some research reported that mean value of WBC and LYM were 8.08–12.75 (10^9^/L) and 2.67–4.77 (10^9^/L) of weaning Hanwoo and Holstein calves [31,32,33]. In the present study, mean value of WBC was 28.55 (10^9^/L) and LYM was 4.83 (10^9^/L) for calves fed RM, which were much higher than that of the research mentioned above. However, mean value of WBC and LYM for calves offered ARM and AFM were very close to the values of the research we have reviewed. Therefore, it can be concluded that the calves offered RM were susceptibility to infection, which may result in detrimental effects on calf health.

## 5. Conclusions

Results from this research indicated that calves fed acidified reconstituted milk or acidified fresh milk had greater average daily gain and withers height growth, lower diarrhea rate, and white blood cell and lymphocytes counts. We concluded that calves offered reconstituted milk were susceptible to infection, which may have a negative effect on calf health. Acidification of reconstituted milk as well as fresh milk had positive effects on growth performance and health status of the preweaning calves.

## Figures and Tables

**Table 1 animals-09-00778-t001:** Composition of different types of milk.

Items	ARM	RM	AFM
Milk lactose, %	4.08	4.48	4.63
Milk fat, %	4.30	4.55	4.74
Milk protein, %	2.86	2.94	3.13
Total solids, %	11.53	12.29	12.84
pH	4.26	6.87	4.32

ARM: acidified reconstituted milk. RM: reconstituted milk. AFM: acidified milk made by using fresh milk.

**Table 2 animals-09-00778-t002:** Bacterial numbers of different types of milk.

Item	Diet			SEM	*p*-Value
	ARM	RM	AFM		
Total Bacterial Count (10^4^ cfu/mL)	43.67 ^a^	322.33 ^b^	38.00 ^a^	44.6	0.001
Lactobacillus (10^3^ cfu/mL)	114.33	52.17	115.00	30.4	0.137
Escherichia coli (10^1^ cfu/mL)	154.83 ^a^	458.67 ^b^	155.67 ^a^	39.1	0.000

ARM: acidified reconstituted milk. RM: reconstituted milk. AFM: acidified milk made by using fresh milk. ^a b^ Mean values within a row with different superscript letter differ significantly (*p* < 0.05). SEM: Standard Error of Mean.

**Table 3 animals-09-00778-t003:** Composition and nutrient levels of the starter (DM basis, %).

Ingredient	Content	Nutrient Levels	Content
Corn	33.00	NEL (Mcal/kg) ^3^	1.69
Soybean meal	23.50	DM ^4^	88.07
Expanded corn	10.00	CP ^5^	20.95
DDGS ^1^	8.00	EE ^6^	3.31
Corn husk	5.07	NDF ^7^	16.34
Extruded soybean	4.00	ADF ^8^	5.48
Corn germ meal	9.00	Ash	7.40
Limestone	1.50	Ca ^9^	0.95
MDCP ^2^	0.15	P ^10^	0.51
NaCl	0.77		
Beer yeast	2.00		
Elancoban	0.01		
Glucose	2.00		
Premix ^11^	1.00		

^1^ DDGS: Distillers Dried Grains with Solubles; ^2^ MDCP:mono-calcium and di-calcium phosphate; ^3^ NEL: net energy for lactation; ^4^ DM: dry matter; ^5^ CP: crude protein; ^6^ EE: ether extract; ^7^ NDF: neutral detergent fiber; ^8^ ADF: acid detergent fibre; ^9^ Ca: Calcium; ^10^ P: phosphorus; ^11^ The premix provided the following per kilogram of the diet: V_A_ 12,000 IU, V_D_ 4000 IU, V_E_ 2900IU, Cu 5000 mg, Fe 9000 mg, Mn 6000 mg, Se 67 mg, I 227 mg, Co 20 mg, Mg 9800 mg.

**Table 4 animals-09-00778-t004:** Growth performance of dairy calves fed different types of milk.

Item	Diet			SEM	*p*-Value
	ARM	RM	AFM		
Initial BW (kg)	67.60	67.60	67.20	2.33	0.513
Final BW (kg)	104.40	102.20	101.80	3.79	0.766
ADG (kg/day)	0.92	0.88	0.96	0.25	0.952
Milk intake (L/day)	11.28	11.17	11.41	0.25	0.637
Milk intake (kg/day DM)	1.34 ^a^	1.41 ^b^	1.51 ^c^	0.03	0.002
Starter intake (kg/day DM)	0.43	0.40	0.42	0.08	0.994
Total feed intake (kg/day DM)	1.76 ^a^	1.81 ^a^	1.92 ^b^	0.03	0.002
Feed efficiency (kg intake/kg gain)	1.85 ^a^	2.05 ^b^	2.00 ^b^	0.03	0.000

ARM: acidified reconstituted milk. RM: reconstituted milk. AFM: acidified milk made by using fresh milk. ^a b c^ Mean values within a row with different superscript letter differ significantly (*p* < 0.05). SEM: Standard Error of Mean.

**Table 5 animals-09-00778-t005:** Body measurements of dairy calves fed different types of milk.

Item	Diet			SEM	*p*-Value
	ARM	RM	AFM		
Initial					
Body length (cm)	83.80	83.60	82.60	0.76	0.275
Withers height (cm)	83.60	83.80	83.00	0.87	0.634
Heart girth (cm)	94.00	93.60	92.20	1.37	0.411
Shin circumference (cm)	12.00	11.80	11.60	0.26	0.335
Final					
Body length (cm)	94.60	94.80	93.80	1.17	0.671
Withers height (cm)	95.20 ^a^	91.00 ^b^	94.60 ^a^	1.44	0.026
Heart girth (cm)	107.20 ^a^	108.00 ^a^	103.00 ^b^	1.78	0.034
Shin circumference (cm)	12.70	12.80	12.40	0.44	0.649
Growth					
Body length (cm)	10.8	11.2	11.2	1.22	0.931
Withers height (cm)	11.6 ^a^	7.20 ^b^	11.6 ^a^	1.45	0.015
Heart girth (cm)	13.2	14.4	10.80	1.76	0.157
Shin circumference (cm)	0.70	1.00	0.80	0.29	0.597

ARM: acidified reconstituted milk. RM: reconstituted milk. AFM: acidified milk made by using fresh milk. ^a b^ Mean values within a row with different superscript letter differ significantly (*p* < 0.05). SEM: Standard Error of Mean.

**Table 6 animals-09-00778-t006:** Fecal score and diarrhea rate of dairy calves fed different types of milk.

Item	Diet			SEM	*p*-Value
	ARM	RM	AFM		
Fecal consistency score					
30–50 day	1.12 ^a^	1.48 ^b^	1.21 ^a^	0.10	0.003
51–60 day	1.20	1.45	1.30	0.13	0.207
61–70 day	1.29 ^a^	1.82 ^b^	1.63 ^b^	0.15	0.006
Diarrhea rate (%)					
30–50 day	0.72 ^a^	11.07 ^b^	0.74 ^a^	0.03	0.000
51–60 day	0.17 ^a^	10.87 ^b^	0.23 ^a^	0.20	0.000
61–70 day	2.90 ^a^	9.40 ^b^	2.48 ^a^	0.41	0.000

ARM: acidified reconstituted milk. RM: reconstituted milk. AFM: acidified milk made by using fresh milk. ^a b^ Mean values within a row with different superscript letter differ significantly (*p* < 0.05). SEM: Standard Error of Mean.

**Table 7 animals-09-00778-t007:** Hematological parameters of dairy calves fed different types of milk.

Item	Diet			SEM	*p*-Value
	ARM	RM	AFM		
Initial					
WBC ^1^ (10^9^/L)	10.57	11.06	10.81	0.98	0.887
LYM ^2^ (10^9^/L)	3.29	5.04	3.38	1.62	0.496
MO ^3^ (10^9^/L)	1.69	1.70	2.57	0.51	0.185
RBC ^4^ (10^12^/L)	4.49	4.69	4.57	0.48	0.921
Hb ^5^ (g/L)	130.60	135.60	131.40	5.21	0.601
HCT ^6^ ( L/L)	0.22	0.23	0.22	0.01	0.383
Final					
WBC (10^9^/L)	12.17 ^a^	28.55 ^b^	12.52 ^a^	4.88	0.008
LYM (10^9^/L)	3.21 ^a^	4.83 ^b^	3.00 ^a^	0.71	0.049
MO (10^9^/L)	1.28	1.36	1.26	0.31	0.947
RBC (10^12^/L)	5.52	6.11	5.78	0.40	0.365
Hb (g/L)	116.20	121.40	118.00	4.61	0.537
HCT (L/L)	0.23	0.26	0.25	0.02	0.279

ARM: acidified reconstituted milk. RM: reconstituted milk. AFM: acidified milk made by using fresh milk. ^a b^ Mean values within a row with different superscript letter differ significantly (*p* < 0.05). ^1^ WBC: white blood cell; ^2^ LYM: lymphocytes; ^3^ MO: monocytes; ^4^ RBC: red blood cell; ^5^ Hb: hemoglobin; ^6^ HCT: hematocrit. SEM: Standard Error of Mean.

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
