# Peer review of "Comparison of Reconstituted, Acidified Reconstituted Milk or Acidified Fresh Milk on Growth Performance, Diarrhea Rate, and Hematological Parameters in Preweaning Dairy Calves"

_animals, 2019, doi:10.3390/ani9100778_

Round 1

Reviewer 1 Report

The manuscript provides interesting information on milk feeding practices and effects on growth and health.  The results are useful and show that if feeding milk that is left out for several hours when feeding high volumes of milk that acidification is a useful method to restrict bacterial growth.  It would have been useful to have a pasteurized fresh milk treatment without acidification to see if this would be useful in that circumstance, however the current study provides valuable comparison and information especially in regard to feeding reconstituted milk with or without acidification. 

There are some missing pieces of information related to the milk feeding and milk intakes.  The methods state that there were 2 feedings per day with 6 L per feeding, but you say that they were fed for ad-libitum intakes.  Please include the as fed amount in liters that the calves ate so the reader is assured that calves actually had ad-libitum intakes.  Was the fresh milk pasteurized before feeding?  Was the fresh milk from the general herd or was it from cows treated with mastitis or that recently calved?  Also, did you measure bacteria concentrations of the milk at feeding and at different times to determine if this would be a main cause of diarrhea between treatments?

For the feed efficiency results, the ARM treatment actually had the greatest efficiency with it needing the least feed per unit of gain compared to the other treatments.  You state that ARM had the lowest efficiency which would make the reader think it was worse with it actually being better.

Table 2: make sure to define abbreviations (DDGS and MDCP) in all tables

Table 3: please include units for feed efficiency (kg intake/kg gain)

Table 5:  include units for the diarrhea rate.  assume it is percentage but the reader may not know

Table 6: make sure to define abbreviations for blood parameters in the footnotes

Reviewer 2 Report

This study by Li et al. provides information that the application of acidified reconstituted milk in preweaning dairy calves. The study shows a new way to use reconstituted milk, which could have positive effects on growth performance and health status of the preweaning dairy calves. the work is well done and the paper is well organized. I have reviewed the paper carefully and the paper can be accepted after minor revision. The following instructions are given to the authors to modify the paper.

Many sentences are difficult to understand. Line 137-138, 192-193 Line 35 The key words ‘acidified milk’ are not be seen in the abstract. Please check and add proper key words. The experiment designed that dairy calves were offered constant milk intake (6 L/d) of the three treatment groups, why the authors discuss the effects of milk intake (DM)? Authors should indicate that whether the fresh milk is antibiotic milk. Line 107 Data were evaluated by one-way analysis of variance Abbreviation

Line 180-181 Please delete the repetitive abbreviation.

Line 197-198 …had greater ADG and withers height growth…

Besides, authors should check the use of abbreviations in both abstract and main body.

Line 160-161 please add the relative references you mentioned. Others

Line 26 Use [(67.40 ±4.08) kg] instead of (67.40 ±4.08) kg

Line 32 Withers height growth of different treatments are significant in the results, it should be stated in the abstract

Line 39 “dairy calves” should be better replaced by “female dairy calves”

Line 43  Replace words “reared on” with “reared by”

Line 64 “1 month of age” can be displayed as “30 days”,

Line 65 Use  [(67.40 ±4.08) kg] instead of (67.40 ±4.08) kg

Line 68 Use space in 40℃

Line 70 Use space in 10℃

Line 71 Use space in 10℃, and how many mL of milk sample was collected for analyzing milk composition, it should be stated clearly.

Line 76  Use space in 6L and add “were” before the word “offered”

Line 89  Please write the formula to calculate the feed efficiency.

Line 91  Were the body measurements tested for 3 consecutive days?

Line 122  The results were not written clearly. Withers height growth and final withers height should be written separately.

Line 126  The unit of the data should not be write after the headline of the table, it should be better put in the Table.

Reviewer 3 Report

The article "Comparison of reconstituted milk, acidified reconstituted milk or acidified fresh milk on growth performance, diarrhea rate and hematological parameters of preweaning dairy calves" by Lingyan Li et al. presents significant leaks, the most important is the statistical analysis. For this reason the data presented in the tables are difficult to understand, because of the differences between the results and the discussion. I suggest to reject the paper.

Round 2

Reviewer 3 Report

Despite the “statistical analysis” description has been improved by the authors, the paper requires other changed before consider it suitable for publication.

Title: I suggest to change the title in: Comparison of reconstituted, acidified and reconstituted milk or acidified fresh milk on growth, performance,diarrhea rate and hematological parameters in preweaning dairy calves

Abstract: check the initial body weight, considering the data reported in table 4, the initial body weight seems different. In addition, you declare that the animals had 1 month of age, but I don’t think that they were born in the some day, so you can add also days and standard deviation of age.

Line 86: have you considered a period of adaptation of the animals before proceeding to collect the data?

Line 194: some researcher?! Please, add the appropriate references.

Lines 216-219: please add a reference in order to support this sentence.

Lines 219-222: please add a reference in order to support this sentence.

Lines 224-226: please move this sentence in the conclusion section.

Group RM showed highest WBC and LYM but the values are in the physiological range for this specie. For this reason, the discussion concerning these results are a speculation. I suggest to remodel the period taking this into account.

Conclusion: I suggest to improve the conclusion in order to understand how your paper address important information to the literature. I suggest also to not use all acronyms in order to better understand the results of the paper.

Tables: I suggest to remove the standard deviation and report the SEM in all the tables. In addition, specify under the table in full “SEM”.

Same references in the text appear different from the guidelines of the journal. Please check it.
